# EarlyBERT: Efficient BERT Training via Early-bird Lottery Tickets

## Abstract

Deep, heavily overparameterized language models such as BERT, XLNet and T5 have achieved impressive success in many natural language processing (NLP) tasks. However, their high model complexity requires enormous computation resources and extremely long training time for both pre-training and fine-tuning. Many works have studied model compression on large NLP models, but only focusing on reducing inference time while still requiring expensive training process. Other works use extremely large batch sizes to shorten the pre-training time, at the expense of higher computational resource demands. In this paper, inspired by the *Early-Bird Lottery Tickets* recently studied for computer vision tasks, we propose EarlyBERT, a general computationally-efficient training algorithm applicable to both pre-training and fine-tuning of large-scale language models. By slimming the self-attention and fully-connected sub-layers inside a transformer, we are the first to identify *structured* winning tickets in the early stage of BERT training. We apply those tickets towards efficient BERT training, and conduct comprehensive pre-training and fine-tuning experiments on GLUE and SQuAD downstream tasks. Our results show that EarlyBERT achieves comparable performance to standard BERT, with 35∼45% less training time.

## 1 Introduction

Large-scale pre-trained language models (*e.g.,* BERT (Devlin et al., 2018), XLNet (Yang et al., 2019), T5 (Raffel et al., 2019)) have significantly advanced the state of the art in the NLP field. Despite impressive empirical success, their computational inefficiency has become an acute drawback in practice. As more and more transformer layers are stacked with larger self-attention blocks, model complexity increases rapidly. For example, compared to BERT-Large model with 340 million parameters, T5 has more than 10 billion to learn. Such high model complexity calls for expensive computational resources and extremely long training time.

Model compression is one approach to alleviating this issue. Recently, many methods propose to encode large NLP models compactly (Sun et al., 2019; Sanh et al., 2019; Sun et al., 2020). However, the focus is solely on reducing computational resources or inference time, leaving the process of searching for the right compact model ever more costly. Furthermore, almost all model compression methods start with a large pre-trained model, which in practice may not exist. Recent work (You et al., 2020b) proposes to use large training batches, which significantly shortens pre-training time of BERT-Large model but demands daunting computing resources (1,024 TPUv3 chips).

In contrast, our quest is to find a general resource-*efficient* training algorithm for large NLP models, which can be applied to both pre-training and fine-tuning stages. Our goal is to trim down the training time, but also avoiding more costs of the total training resources (*e.g.*, taking large-batch or distributed training). To meet this challenge demand, we draw inspirations from a recent work (You et al., 2020a) that explores the use of Lottery Ticket Hypothesis (LTH) for efficient training of computer vision models. LTH was first proposed in Frankle & Carbin (2019) as an exploration to understand the training process of deep networks. The original LTH substantiates a trainable sparse sub-network at initialization, but it cannot be directly utilized for efficient training, since the sub-network itself has to be searched through a tedious iterative process. In addition, most LTH works discussed only unstructured sparsity. The study of You et al. (2020a) presents new discoveries that structured lottery tickets can emerge in early stage of training (*i.e.*, Early-Bird Ticket), and there-

fore a structurally sparse sub-network can be identified with much lower costs, leading to practical efficient training algorithms.

Inspired by the success of LTH and Early-Bird Ticket, we propose EarlyBERT, a general efficient training algorithm based on structured Early-Bird Tickets. Due to the vast differences between the architectures and building blocks of computer vision models and BERT, directly extending the method of (You et al., 2020a) is not applicable to our work. By instead using network slimming (Liu et al., 2017) on the self-attention and fully-connected sub-layers inside a transformer, we are the first to introduce an effective approach that can identify *structured winning tickets in the early stage of BERT training*, that are successfully applied for efficient language modeling pre-training and fine-tuning. Extensive experiments on BERT demonstrate that EarlyBERT can save 35~45% training time without sacrificing accuracy, when evaluated on GLUE and SQuAD benchmarks.

## 2 RELATED WORK

**Efficient NLP Models** It is well believed that BERT and other large NLP models are considerably overparameterized (McCarley, 2019; Sun et al., 2019). This explains the emergence of many model compression works, which can be roughly categorized into quantization (Shen et al., 2020; Zafrir et al., 2019), knowledge distillation (Sun et al., 2019; Jiao et al., 2019; Sanh et al., 2019; Sun et al., 2020), dynamic routing (Fan et al., 2019; Xin et al., 2020), and pruning (Li et al., 2020; Wang et al., 2019; McCarley, 2019; Michel et al., 2019). Almost all model compression methods focus on reducing inference time, while their common drawback is the reliance on fully-trained and heavily-engineered dense models, before proceeding to their compact, sparse versions - which essentially transplants the resource burden from the inference to the training stage

Pruning is the mainstream approach for compressing BERT so far. McCarley (2019) proposed to greedily and iteratively prune away attention heads contributing less to the model. Wang et al. (2019) proposed to structurally prune BERT models using low-rank factorization and augmented Lagrangian $\ell_0$ norm regularization. McCarley (2019) pruned less important self-attention heads and slices of MLP layers by applying $\ell_0$ regularization to the coefficient corresponding to each head/MLP layer. Another line of works aim to reduce the training time of transformer-based models via large-batch training and GPU model parallelism (You et al., 2020b; Shoeybi et al., 2019). Our work is orthogonal to those works, and can be readily combined for further efficiency boost.

**Lottery Ticket Hypothesis in Computer Vision** Lottery Ticket Hypothesis (LTH) was firstly proposed in Frankle & Carbin (2019), which shed light on the existence of sparse sub-networks (*i.e.*, winning tickets) at initialization with non-trivial sparsity ratio that can achieve almost the same performance (compared to the full model) when trained alone. The winning tickets are identified by pruning fully trained networks using the so-called Iterative Magnitude-based Pruning (IMP). However, IMP is expensive due to its iterative nature. Moreover, IMP leads to unstructured sparsity, which is known to be insufficient in reducing training cost or accelerating training speed practically. Those barriers prevent LTH from becoming immediately helpful towards efficient training. Morcos et al. (2019) studies the transferability of winning tickets between datasets and optimizers. Zhou et al. (2019) investigates different components in LTH and observes the existence of super-masks in winning tickets. Lately, You et al. (2020a) pioneers to identify Early-Bird Tickets, which emerge at the early stage of the training process, and contain structured sparsity when pruned with Network Slimming (Liu et al., 2017). Early-bird tickets mitigate the two limitations of IMP aforementioned, and renders it possible to training deep models efficiently, by drawing such tickets early in the training and then focusing on training this compact subnetwork only.

**Lottery Ticket Hypothesis in NLP** All above works evaluate their methods on computer vision models. For NLP models, previous work has also found that matching subnetworks exist in training on Transformers and LSTMs (Yu et al., 2019; Renda et al., 2020). Evci et al. (2020) derived an algorithm for training sparse neural networks according to LTH and applied it to a character-level language modeling on WikiText-103. For BERT models, a latest work (Chen et al., 2020) found that the pre-trained BERT models contain sparse subnetworks, found by unstructured IMP at 40% to 90% sparsity, that are independently trainable and transferable to a range of downstream tasks with no performance degradation. Another concurrent work (Prasanna et al., 2020) aims to find structurally sparse lottery tickets for BERT, by pruning entire attention heads and MLP layers. Their experiments turn out that all subnetworks ("good" and "bad") have "comparable performance" when fined-tuned on downstream tasks, leading to their "all tickets are winning" conclusion.

Nevertheless, both works (Chen et al., 2020; Prasanna et al., 2020) examine only the pre-trained BERT model, *i.e.*, finding tickets with regard to the fine-tuning stage on downstream tasks. To our best knowledge, no existing study analyzes the LTH at the pre-training stage of BERT; nor has any work discussed the efficient BERT training using LTH, for either pre-training or fine-tuning stage. In comparision, our work represents the first attempt of introducing LTH to both efficient pre-training and efficient fine-tuning of BERT. Our results also provide positive evidence that LTH and Early-Bird Tickets in NLP models are amendable to structured pruning too.

## 3 THE EARLYBERT FRAMEWORK

In this section, we first revisit the original Lottery Ticket Hypothesis (LTH) (Frankle & Carbin, 2019) and its variant Early-Bird Ticket (You et al., 2020a), then describe our proposed EarlyBERT.

### 3.1 REVISITING LOTTERY TICKET HYPOTHESIS

Denote $f(x; \theta)$ as a deep network parameterized by $\theta$ and $x$ as its input. A sub-network of $f$ can be characterized by a binary mask $m$, which has exactly the same dimension as $\theta$. When applying the mask $m$ to the network, we obtain the sub-network $f(x; \theta \odot m)$, where $\odot$ is the Hadamard product operator. LTH states that, for a network initialized with $\theta_0$, an algorithm called Iterative Magnitude Pruning (IMP) can identify a mask $m$ such that the sub-network $f(x; \theta_0 \odot m)$ can be trained to have no worse performance than the full model $f$ following the same training protocol. Such a sub-network $f(x; \theta_0 \odot m)$, including both the mask $m$ and initial parameters $\theta_0$, is called a *winning ticket*. The IMP algorithm works as follows: (1) initialize $m$ as an all-one mask; (2) fully train $f(x; \theta_0 \odot m)$ to obtain a well-trained $\theta$; (3) remove a small portion of weights with the smallest magnitudes from $\theta \odot m$ and update $m$; (4) repeat (2)-(3) until a certain sparsity ratio is achieved.

Two obstacles prevent LTH from being directly applied to efficient training. First, the iterative process in IMP is essential to preserve the performance of LTH; however, this is computationally expensive, especially when the number of iterations is high. Second, the original LTH does not pursue any structured sparsity in the winning tickets. In practice, unstructured sparsity is difficult to be utilized for computation acceleration even when the sparsity ratio is high (Wen et al., 2016).

To mitigate these gaps, Early-Bird Tickets are proposed by You et al. (2020a), who discovers that when using structured mask $m$ and a properly selected learning rate, the mask $m$ quickly converges and the corresponding mask emerges as the winning ticket in the early stage of training. The early emergence of winning tickets and the structured sparsity are both helpful in reducing computational cost in the training that follows. You et al. (2020a) focuses on computer vision tasks with convolutional networks such as VGG (Simonyan & Zisserman, 2014) and ResNet (He et al., 2016). Inspired by this, we set out to explore whether there are structured winning tickets in the early stage of BERT training that can significantly accelerate language model pre-training and fine-tuning.

### 3.2 DISCOVERING EARLYBERT

The proposed EarlyBERT[1] training framework consists of three steps: $(i)$ *Searching Stage*: jointly train BERT and the sparsity-inducing coefficients to be used to draw the winning ticket; $(ii)$ *Ticket-drawing Stage*: draw the winning ticket using the learned coefficients; and $(iii)$ *Efficient-training Stage*: train EarlyBERT for pre-training or downstream fine-tuning.

**Searching Stage** To search for the key sub-structure in BERT, we follow the main idea of Network Slimming (NS) (Liu et al., 2017). However, pruning in NS is based on the scaling factor $\gamma$ in batch normalization, which is not used in most NLP models such as BERT. Therefore, we make necessary modifications to the original NS so that it can be adapted to pruning BERT. Specifically, we propose to associate attention heads and intermediate layers of the fully-connected sub-layers in a transformer with learnable coefficients, which will be jointly trained with BERT but with an additional $\ell_1$ regularization to promote sparsity.

---

[1]EarlyBERT refers to the winning ticket discovered by the proposed 3-stage framework, which is equivalent to the resulting pruned BERT model after drawing the winning ticket. We also interchangeably use EarlyBERT as the name of the proposed framework.

Some studies (Michel et al., 2019; Voita et al., 2019) find that the multi-head self-attention module of transformer can be redundant sometimes, presenting the possibility of pruning some heads from each layer of BERT without hurting model capacity. A multi-head attention module (Vaswani et al., 2017) is formulated as:

$$\text{MultiHead}(Q, K, V) = \text{Concat}(\text{head}_1, \ldots, \text{head}_h)W^O \tag{1}$$

$$\text{where } \text{head}_i = \text{Attention}(QW_i^Q, KW_i^K, VW_i^V), \tag{2}$$

where the projections $W^O, W_i^Q, W_i^K, W_i^V$ are used for output, query, key and value. Inspired by Liu et al. (2017), we introduce a set of scalar coefficients $c_i^h$ ($i$ is the index of attention heads and $h$ means "head") inside $\text{head}_i$:

$$\text{head}_i = c_i^h \cdot \text{Attention}(QW_i^Q, KW_i^K, VW_i^V). \tag{3}$$

After the self-attention sub-layer in each transformer layer, the output $\text{MultiHead}(Q, K, V)$ will be fed into a two-layer fully-connected network, in which the first layer increases the dimension of the embedding by 4 times and then reduces it back to the hidden size (768 for BERT$_{\text{BASE}}$ and 1,024 for BERT$_{\text{LARGE}}$). We multiply learnable coefficients to the intermediate neurons:

$$\text{FFN}(x) = c^f \cdot \max(0, xW_1 + b_1)W_2 + b_2. \tag{4}$$

These modifications allow us to jointly train BERT with the coefficients, using the following loss:

$$\mathcal{L}(f(\cdot; \theta), c) = \mathcal{L}_0(f(\cdot; \theta), c) + \lambda \|c\|_1, \tag{5}$$

where $\mathcal{L}_0$ is the original loss function used in pre-training or fine-tuning, $c$ is the concatenation of all the coefficients in the model including those for attention heads and intermediate neurons, and $\lambda$ is the hyper-parameter that controls the strength of regularization.

Note that in this step, the joint training of BERT and the coefficients are still as expensive as normal BERT training. However, the *winning strategy* of EarlyBERT is that we only need to perform this joint training for a few steps, before the winning ticket emerges, which is much shorter than the full training process of pre-training or fine-tuning. In other words, we can identify the winning tickets at a very low cost compared to the full training. Then, we draw the ticket (*i.e.*, the EarlyBERT), reset the parameters and train EarlyBERT that is computationally efficient thanks to its structured sparsity. Next, we introduce how we draw EarlyBERT from the learned coefficients.

**Ticket-drawing Stage**    After training BERT and coefficients $c$ jointly, we draw EarlyBERT using the learned coefficients with a magnitude-based metric. Note that we prune attention heads and intermediate neurons separately, as they play different roles in BERT.

We prune the attention heads whose coefficients have the smallest magnitudes, and remove them from the computation graph. We also prune the rows in $W^O$ (see Eqn. (1)) that correspond to the removed heads. Note that this presents a design choice: should we prune the heads *globally* or *layer-wisely*? In this paper, we use layer-wise pruning for attention heads, because the number of heads in each layer is very small (12 for BERT$_{\text{BASE}}$ and 16 for BERT$_{\text{LARGE}}$). We observe empirically that if pruned globally, the attention heads in some layers may be completely removed, making the network un-trainable. Furthermore, Ramsauer et al. (2020) observes that attention heads in different layers exhibit different behaviors. This also motivates us to only compare importance of attention heads within each layer.

Similar to pruning attention heads, we prune intermediate neurons in the fully-connected sub-layers. Pruning neurons is equivalent to reducing the size of intermediate layers, which leads to a reduced size of the weight matrices $W_1$ and $W_2$ in Eqn. (4). Between global and layer-wise pruning, empirical analysis shows that global pruning works better. We also observe that our algorithm naturally prunes more neurons for the later layers than earlier ones, which coincides with many pruning works on vision tasks. We leave the analysis of this phenomenon as future work.

**Efficient-training Stage**    We then train EarlyBERT that we have drawn for pre-training or fine-tuning depending on the target task. If we apply EarlyBERT to pre-training, the initialization $\theta_0$ of BERT will be a random initialization, the same setting as the original LTH (Frankle & Carbin, 2019)

and Early-Bird Tickets (You et al., 2020a). If we apply EarlyBERT to fine-tuning, then $\theta_0$ can be any pre-trained model. We can also moderately reduce the training steps in this stage without sacrificing performance, which is empirically supported by the findings in Frankle & Carbin (2019); You et al. (2020a) that the winning tickets can be trained more effectively than the full model. In practice, the learning rate can also be increased to speed up training, in addition to reducing training steps.

Different from unstructured pruning used in LTH and many other compression works (Frankle & Carbin, 2019; Chen et al., 2020), structurally pruning attention heads and intermediate neurons in fully-connected layers can directly reduce the number of computations required in the transformer layer, and shrink the matrix size of the corresponding operations, yielding a direct reduction in computation and memory costs.

## 3.3 VALIDATION OF EARLYBERT

**Early Emergence**  Following a similar manner in You et al. (2020a), we visualize the normalized mask distance between different training steps, to validate the early emergence of winning tickets. In Figure 1, the axes in the plots are the number of training steps finished. We only use one fully-connected sub-layer to plot Figure 1(b),1(d) due to high dimensionality. In both pre-training and fine-tuning, the mask converges in a very early stage of the whole training process. Although we observe an increase of mask distance in fully-connected layers during pre-training (in Figure 1(b)), this can be easily eliminated by early stopping and using mask distance as the exit criterion.

**Non-trivial Sub-network**  Here, by *non-trivial* we mean that with the same sparsity ratio as in EarlyBERT, randomly pruned model suffers from significant performance drop. The performance drop happens even if we only prune attention heads. We verify this by running fine-tuning experiments on $\text{BERT}_{\text{BASE}}$. Specifically, we prune 4 heads from each transformer layer in $\text{BERT}_{\text{BASE}}$ and EarlyBERT. We fine-tune $\text{BERT}_{\text{BASE}}$ for 3 epochs with an initial learning rate $2 \times 10^{-5}$. We run the searching stage for 0.2 epochs with $\lambda = 1 \times 10^{-4}$, draw EarlyBERT with pruning ratio $\rho = 1/3$, and then fine-tune EarlyBERT for 2 epochs with doubled initial learning rate. For the randomly pruned models, we randomly prune 4 heads in each layer and follow the same fine-tuning protocol as EarlyBERT. The reported results of randomly pruned models are the average of 5 trials with different seeds for pruning. The results on three tasks from GLUE benchmark (Wang et al., 2018) presented in Table 1 show that randomly pruned model consistently under-performs EarlyBERT with a significant gap, supporting our claim that EarlyBERT indeed identifies non-trivial sub-structures.

Table 1: Comparison between randomly-pruned models and EarlyBERT on 4 GLUE tasks. We prune 4 heads in each layer.

| Methods | MNLI | QNLI | QQP | SST-2 |
|---|---|---|---|---|
| $\text{BERT}_{\text{BASE}}$ | 83.16 | 90.59 | 90.34 | 91.70 |
| $\text{EarlyBERT}_{\text{BASE}}$ | 83.58 | 90.33 | 90.41 | 92.09 |
| Random | 82.26 | 88.87 | 90.12 | 91.17 |

## 4 EXPERIMENTS

### 4.1 EXPERIMENTAL SETTING

**Backbone Models**  Following the official BERT implementation (Devlin et al., 2018; Wolf et al., 2019), we use both $\text{BERT}_{\text{BASE}}$ (12 transformer layers, hidden size 768, 3,072 intermediate neurons, 12 self-attention heads per layer, 110M parameters in total) and $\text{BERT}_{\text{LARGE}}$ (24 transformer layers, hidden size 1,024, 4,096 intermediate neurons, 16 self-attention heads per layer, 340M parameters in total) for experiments.

**Datasets**  We use English Wikipedia (2,500M words) as the pre-training data. For fine-tuning experiments and evaluation of models in the pre-training experiments, we use tasks from GLUE benchmark (Wang et al., 2018) and a question-answering dataset SQuAD v1.1 (Rajpurkar et al., 2016). Note that as our goal is efficient pre-training and fine-tuning, we focus on larger datasets from GLUE (MNLI, QNLI, QQP and SST-2), as it is less meaningful to discuss efficient training

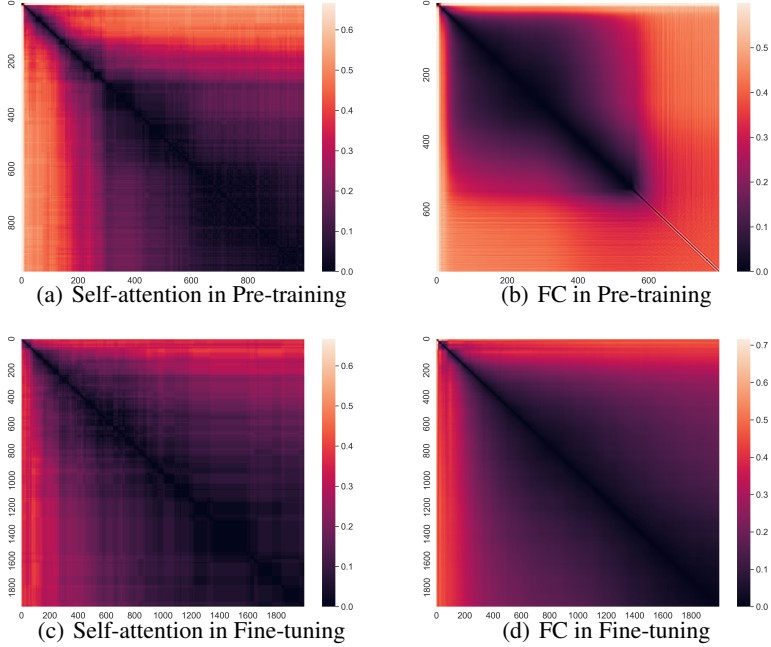

Figure 1: Illustration of Mask Distance. Top: mask distance observed in pre-training. Bottom: mask distance observed in fine-tuning. The color represents the normalized mask distance between different training steps. The darker the color, the smaller the mask distance.

on very small datasets. We use the default training settings for pre-training and fine-tuning on both models. To evaluate model performance, we use Matthew's correlation score for CoLA, matched accuracy for MNLI, F1-score for SQuAD v1.1, and accuracy in percentage for other tasks on GLUE. We omit % symbols in all the tables on accuracy results.

**Implementation Details**    For the vanilla BERT, we fine-tune on GLUE datasets for 3 epochs with initial learning rate $2 \times 10^{-5}$, and for 2 epochs on SQuAD with initial learning rate $3 \times 10^{-5}$; we use AdamW (Loshchilov & Hutter, 2017) optimizer for both cases. For pre-training, we adopt LAMB optimization technique (You et al., 2020b), which involves two phases of training: the first 9/10 of the total training steps uses a sequence length of 128, while the last 1/10 uses a sequence length of 512. Pre-training by default has 8,601 training steps and uses 64k/32k batch sizes and $6 \times 10^{-3}/4 \times 10^{-3}$ initial learning rates for the two phases, respectively. All experiments are run on 16 NVIDIA V100 GPUs.

## 4.2 EXPERIMENTS ON FINE-TUNING

The main results of EarlyBERT in fine-tuning are presented in Table 2. When drawing EarlyBERT, we prune 4 heads in each layer from BERT$_{\text{BASE}}$ and 6 heads from BERT$_{\text{LARGE}}$, and globally prune 40% intermediate neurons in fully-connected sub-layers in both models. We reduce the training epochs to two on GLUE benchmark and scale up the learning rate by 2 to buffer the effect of reduced epochs. For SQuAD dataset, we keep the default setting, as we find SQuAD is more sensitive to the number of training epochs. Ablation studies on the effects of the number of training epochs and learning rate are included in the following section.

Several observations can be drawn from Table 2. *Firstly*, in most tasks, EarlyBERT saves over 40% of the total training time without inducing much performance degradation. It can also outperform another strong baseline LayerDrop (Fan et al., 2019), which drops one third of the layers so that the number of remaining parameters are comparable to ours. Note that LayerDrop models are fine-tuned for three full epochs, yet EarlyBERT is still competitive in most cases. *Secondly*, we consistently observe obvious performance advantage of EarlyBERT over randomly pruned models, which pro-

Table 2: Performance of EarlyBERT (fine-tuning) compared with different baselines.

| Methods | MNLI | QNLI | QQP | SST-2 | SQuAD | Time Saved[2] |
|---|---|---|---|---|---|---|
| BERT$_{\text{BASE}}$ | 83.16 | 90.59 | 90.34 | 91.70 | 87.50 | - |
| EarlyBERT$_{\text{BASE}}$ | 81.81 | 89.18 | 90.06 | 90.71 | 86.13 | 40~45% |
| Random$_{\text{BASE}}$ | 79.92 | 84.46 | 89.42 | 89.68 | 84.47 | 45~50% |
| LayerDrop (Fan et al., 2019) | 81.27 | 88.91 | 88.06 | 89.89 | 84.25 | ~33% |
| BERT$_{\text{LARGE}}$ | 86.59 | 92.29 | 91.59 | 92.21 | 90.76 | - |
| EarlyBERT$_{\text{LARGE}}$ | 85.13 | 89.22 | 90.64 | 90.94 | 89.45 | 35~40% |
| Random$_{\text{LARGE}}$ | 78.45 | 84.46 | 89.89 | 88.65 | 88.79 | 40~45% |
| LayerDrop (Fan et al., 2019) | 85.12 | 91.12 | 88.88 | 89.97 | 89.44 | ~33% |

vides another strong evidence that EarlyBERT does discover nontrivial key sparse structures. Even though there still exists a margin between EarlyBERT and the baseline (like (You et al., 2020a) also observed similarly in their tasks), the existence of structured winning tickets and its potential for efficient training is highly promising. We leave as future work to discover winning tickets of higher sparsity but better quality.

**Ablation Studies on Fine-tuning**  We perform extensive ablation studies to investigate important hyperparameter settings in EarlyBERT, using EarlyBERT$_{\text{BASE}}$ as our testing bed. For all experiments, we use the average accuracy on the larger datasets from GLUE benchmark (MNLI, QNLI, QQP and SST-2) as the evaluation metric.

- **Number of training epochs and learning rate.** We first investigate whether we can properly reduce the number of training epochs, and if scaling the learning rate can help compliment the negative effect caused by reducing training steps. Results in Figure 2 show that when we fine-tune EarlyBERT for fewer epochs on GLUE benchmark, up-scaling learning rate first helps to recover performance, and then causes decrease again. We will use two epochs and $4 \times 10^{-5}$ as learning rate for EarlyBERT on GLUE experiments.

- **Regularization strength** $\lambda$**.** A proper selection of the regularization strength $\lambda$ decides the quality of the winning ticket, consequently the performance of EarlyBERT after pre-training/fine-tuning. Results on different strength settings in Table 3 show that the regularization strength $\lambda$ has marginal influence on EarlyBERT performance. We use $\lambda = 10^{-4}$ that achieves the best performance in following experiments.

- **Pruning ratios** $\rho$**.** We further investigate the effects of different pruning ratios as well as layer-wise/global pruning on the performance of EarlyBERT. As discussed in Sec. 3.2, we only consider layer-wise pruning for self-attention heads. Table 4 shows that the performance monotonically decreases when we prune more self-attention heads from BERT; however, we see a slight increase and then a sharp decrease in accuracy, when the pruning ratio is raised for intermediate neurons in fully-connected sub-layers (40% pruning ratio seems to be the sweet spot). We also observe consistent superiority of global pruning over layer-wise pruning for intermediate neurons.

## 4.3 Experiments on Pre-training

We also conduct pre-training experiments and present the main results in Table 5. Similar to the settings in fine-tuning experiments, we prune 4 heads in each layer from BERT$_{\text{BASE}}$ and 6 heads from BERT$_{\text{LARGE}}$; however, we prune slightly fewer (30%) intermediate neurons in fully-connected sub-layers in both models, since we empirically observe that pre-training is more sensitive to aggressive intermediate neuron pruning. In both phases of pre-training, we reduce the training steps to 80% of the default setting when training EarlyBERT (based on the ablation study shown in Figure 3). Other hyperparameters for pre-training follow the default setting described in Sec. 4.1. All models are fine-tuned and evaluated on GLUE (Wang et al., 2018) and SQuAD v1.1 datasets (Rajpurkar et al., 2016) with the default setting. Since we observe that the randomly pruned models do not competitive performance in fine-tuning experiment for all downstream tasks, in this section we focus on comparing the achievable performance EarlyBERT with full BERT baseline.

---

[2]The time saving during training is roughly estimated using average time per mini-batch and thus is not very accurate and only serves as refernece.

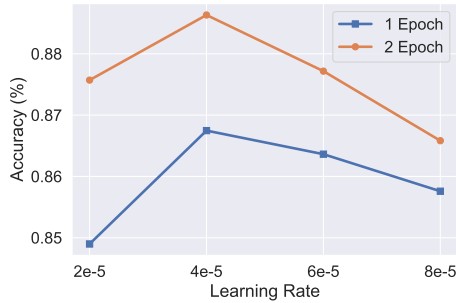

Figure 2: Effect of reducing training epochs and up-scaling learning rate for EarlyBERT in fine-tuning.

Table 3: Ablation of regularization strength $\lambda$.

| $\lambda$ | $10^{-4}$ | $10^{-3}$ | $10^{-2}$ |
|---|---|---|---|
| Avg. Acc. | **89.10** | 88.81 | 88.93 |

Table 4: Ablation of pruning ratios on self-attention heads and intermediate neurons.

| # Pruned Heads | 4 | 5 | 6 |
|---|---|---|---|
| Layer-wise pruning | **89.10** | 88.69 | 88.26 |

| # Pruned Neurons | 30% | 40% | 50% |
|---|---|---|---|
| Layer-wise pruning | 88.33 | **88.48** | 87.91 |
| Global pruning | 88.54 | **88.70** | 88.01 |

Table 5: Performance of EarlyBERT (pre-training) compared with BERT baselines.

| Methods | CoLA | MNLI | MRPC | QNLI | QQP | RTE | SST-2 | SQuAD |
|---|---|---|---|---|---|---|---|---|
| BERT$_{\text{BASE}}$ | 0.45 | 81.40 | 84.07 | 89.86 | 89.80 | 60.29 | 90.48 | 87.60 |
| EarlyBERT$_{\text{BASE}}$ | 0.41 | 79.97 | 80.39 | 89.86 | 89.44 | 61.01 | 90.94 | 85.48 |
| BERT$_{\text{LARGE}}$ | 0.50 | 83.56 | 85.90 | 90.44 | 90.45 | 59.93 | 92.55 | 90.43 |
| EarlyBERT$_{\text{LARGE}}$ | 0.47 | 82.54 | 85.54 | 90.46 | 90.38 | 61.73 | 91.51 | 89.36 |

From the results presented in Table 5, we can see that on downstream tasks with larger datasets such as QNLI, QQP and SST-2 we can achieve accuracies that are close to BERT baseline (within 1% accuracy gaps except for EarlyBERT$_{\text{BASE}}$ on MNLI and SQuAD). However, on downstream tasks with smaller learning rate, the patterns are not consistent: we observe big drops on CoLA and MRPC but improvement on RTE. Overall, EarlyBERT achieves comparable performance while saving 30~35% training time thanks to its structured sparsity and reduction in training steps.

**Reducing Training Steps in Pre-training** We investigate whether EarlyBERT, when non-essential heads and/or intermediate neurons are pruned, can train more efficiently, and whether we can reduce the number of training steps in pre-training. This can further help reduce training cost in addition to the efficiency gain from pruning. We use EarlyBERT$_{\text{BASE}}$-Self (only self-attention heads are pruned when drawing the winning ticket) as the testing bed. Figure 3 shows the performance decreases more when we reduce the number of training steps to 60% or less. Reducing it to 80% seems to be a sweet point with the best balance between performance and efficiency.

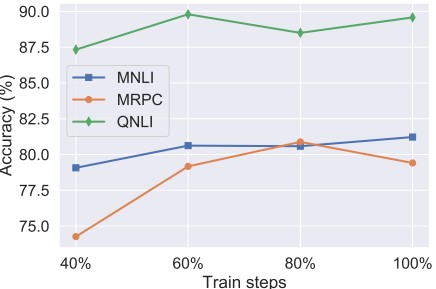

Figure 3: Effect of reducing training steps in pre-training on EarlyBERT$_{\text{BASE}}$.

## 5 CONCLUSION

In this paper, we present EarlyBERT, an efficient training framework for large-scale language model pre-training and fine-tuning. Based on Lottery Ticket Hypothesis, EarlyBERT identifies structured winning tickets in an early stage, then uses the pruned network for efficient training. Experimental results on GLUE and SQuAD demonstrate that the proposed method is able to achieve comparable performance to standard BERT with much less training time. Future work includes applying Early-BERT to other pre-trained language models and exploring more data-efficient strategies to enhance the current training pipeline.

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
