# OpenReview forum: "EarlyBERT: Efficient BERT Training via Early-bird Lottery Tickets"
_ICLR.cc/2021/Conference — Reject_

### Official Review · AnonReviewer1 · 2020-10-24
**Interesting and promising results, but important methodological unclarities.**

**Rating:** 5
**Confidence:** 4

**Review:**

This paper tackles a very important and under-studied problem: reducing the cost of training NLP models. The authors present a method that builds on the lottery ticket hypothesis (LTH). The authors first identify redundant structures early during training, then prune these structures, which leads to faster training. The authors experiment both with pre-training and fine-tuning of contextual models (BERT-{base,large}) and claim large reduction in training time, with reasonable loss in performance.
Despite very encouraging results, several important methodological questions about the source of the efficiency gains and other aspects of the paper are left unanswered. I cannot recommend accepting this paper in its current form, but am looking forward to reading the authors' response which might clarify things.


Detailed comments:

Training (especially pretraining) costs have been going wild in AI and NLP more particularly, which leads to large costs ([1]) as well as potential environmental problems ([2]). Reducing these costs could have a very high impact on the field, allowing many more researchers to participate in state-of-the-art research [3].
As a result, this paper has a great potential, and its results seem very promising.
Nonetheless, the current paper leaves too many open questions regarding the validity of the experiments.
First, much of the improvement (I think) comes from reducing the number of epochs and/or the number of steps. For fine-tuning, the authors run their model for 2.2 epochs, while their baseline model runs for 3 epochs, roughly 30% more which accounts for much of the reduction observed in Table 2. Similarly, for pretraining, the model runs 80% of the training steps (20% reduction), which accounts much of the training time reduction reported on section 4.3. Running a baseline model that runs *for the same amount of time* is essential to appreciate the contribution of this work (e.g., repeat the same analysis in Figure 3 for the vanilla BERT).
Second, the authors argue for a large reduction in runtime, but are very cryptic about how they actually measure this reduction (footnote 2), while reporting number in ranges of 5%. As the main contribution of this paper is the increased efficiency of the proposed approach, it must be clear how efficiency is measured. Finally, writing in general can be made clearer:

1. Last sentence of intro: "without scarifying accuracy" seems like an inaccurate description of the results presented in this paper. around 1% might be reasonable for 30-40% reduction in training time, but it is certainly a reduction in accuracy.
2. Some description of Network sliming would help
3. The term "intermediate neurons" (section 3.2) was unclear to me.
4. Section 3.3: how is mask difference defined?
5. Figure 1 was unclear to me. What do the axis represent? The authors say "the axes in the plots are the number of training steps finished." so why do you need two of them?
6. The "Non-trivial Sub-network" paragraph feels like it should be part of the Experiments section.
7. Implementation details are only given for the vanilla BERT Are they similar to the EarlyBERT model as well?
8. "Since we observe that the randomly pruned models do not competitive performance ...": how uncompetitive? I would have liked to see these results (also, please fix grammar in this sentence)
9. "Reducing it to 80% seems to be a sweet point with the best balance between performance and efficiency." -> I would disagree. The graph indicates that for MNLI and QNLI 60% seems like a better choice.

Questions:
1. "We observe empirically that if pruned globally, the attention heads in some layers may be completely removed, making the network un-trainable.": in this case, couldn't the authors remove a full layer?
2. The difference in the ablation results seem quite small (tables 3 and 4). Are they statistically significant?

Minor:
1. Missing period at the end of the first paragraph in the related work section.
2. Second paragraph of related work: McCarley et al. (2019) appears twice with different descriptions, is this intentional?
3. The authors say "we focus on larger datasets from GLUE (MNLI, QNLI, QQP and SST-2), as it is less meaningful to discuss efficient training", but then report and analyze results from other GLUE datasets as well.
4. "Hon downstream tasks with smaller learning rate" -> Do you mean smaller datasets?





References:
[1] Sharir, O., Peleg, B., and Shoham, Y. (2020). The cost of training NLP models: A concise overview. arXiv:2004.08900.
[2] Strubell, E., Ganesh, A., and McCallum, A. (2019). Energy and policy considerations for deep learning in NLP. In Proc. of ACL.
[3] Schwartz, R., Dodge, J., Smith, N. A., and Etzioni, O. (2019). Green AI. arXiv:1907.10597.

---

> ### Author Response · Authors · 2020-11-18
> **Response to Reviewer 1 (Continued)**
>
> We sincerely thank you for pointing out the ambiguity and unclarity in the writing. We will add details and clarify definitions as suggested in the final version. Specifically,
>
> 1. Thank you for pointing this out. We will tone down the performance of EarlyBERT in the paper.
> 2. We will add more description of network slimming in the final version.
> 3. In each transformer layer, there is one two-layer fully-connected network. We call the output of the first layer after the activation function as the “intermediate neurons”.
> 4. The mask distance is defined as the Hamming distance between two binary masks.
> 5. In every step of the training process (either fine-tuning or pre-training), we can draw the pruning mask given the learned coefficients c. Figure 1 is like a confusion matrix, the value of position (i, j) represents the mask distance between the pruning masks drawn at step i and step j.
> 6. The “Non-trivial Sub-networks” part is a quick proof of concept that EarlyBERT does not find some trivial sub-networks. We will consider relocating this part of content.
> 7. The model definition of EarlyBERT is similar to vanilla BERT models. The training and pruning hyperparameters are studied in Section 4.2. We will make the model and training implementation details more clear in the final version.
> 8. In "Since we observe that the randomly pruned models do not have competitive performance ...", we mean as we observed in the fine-tuning experiments, the randomly pruned models have inferior performance compared to EarlyBERT. The corresponding results are shown in Table 1. We will fix the grammar in the updated version.
> 9. We first apologize that we reported the wrong MNLI accuracy with 60% training steps, which should be 80.10%, inferior to 80.58% with 80% training steps. We will correct this in the final version. A full comparison is shown in the table below. We can see that 80% training steps is consistently better than 60% training steps with the only exception on QNLI. Hence, we think 80% training steps is a better trade-off between training cost and performance. We will clarify this in the final version.
>
> |Training steps | CoLA | MNLI    | MRPC   | QNLI     | QQP     | RTE      | SST-2   |
> |------------------- |:-------:|:----------:|:----------:|:-----------:|:----------:|:----------:|:----------:|
> | 80%               | 0.47   | 80.58% | 80.88% | 88.50% | 89.64% | 64.98% | 90.60% |
> | 60%               | 0.37   | 80.10% | 77.94% | 89.64% | 89.55% | 63.90% | 90.14% |
>
> Minor points:
> 1. We will add the missing period.
> 2. This is a mistake. We will fix the citation in related work in the final version.
> 3. When we apply EarlyBERT to fine-tuning only, training efficiency will only make sense when we fine-tune on large tasks in GLUE. This is why we only report MNLI, QNLI, QQP, SST-2 and SQuAD in Table 2. However, when EarlyBERT is applied to pre-training, the training time is dominated by the pre-training stage. In this case, we reported the performance on all tasks on GLUE in Table 5. We will make this clear.
> 4. Yes, we mean “smaller datasets”. Thank you for pointing out. We will fix it in the final version.
>
> We hope that you can raise your score if you find our answers address your questions. Thank you!

---

> ### Author Response · Authors · 2020-11-18
> **Response to Reviewer 1**
>
> Thank you for your insightful comments. Below, we provide detailed responses to your concerns.
>
> Firstly, we would like to highlight that EarlyBERT saves more than 25% per-step training computation time besides the reduction of training steps with the settings used in the paper (4 heads pruned per-layer and 40% FC neurons pruned globally). This computational efficiency is measured following the strict protocol described in the following paragraph. Moreover, we repeat the same analysis in Figure 3 for vanilla BERT as suggested. The results are shown in the table below. We can see that on tasks with fairly large dataset (e.g. SQuAD, MNLI, QNLI, QQP and SST-2) the results are relatively consistent: reducing the training steps will hurt the performance of BERT on downstream tasks but with acceptable degradation using 80% and 60% training steps. More obvious accuracy drops will be observed when we reduce the training steps to 40% of the default aggressively. Results on smaller tasks are not very stable, which calls for multiple runs of experiments to exclude the influence of randomness. We will include this analysis of the reduction in training steps for BERT and EarlyBERT in the updated version.
>
> | Training Steps | SQuAD | CoLA | MNLI | MRPC | QNLI | QQP | RTE | SST-2 |
> |:--------------|:-----:|:----:|:------:|:------:|:------:|:------:|:------:|:------:|
> |      100%      | 87.60 | 0.45 | 81.40% | 84.07% | 89.86% | 89.80% | 60.29% | 90.48% |
> |       80%      | 87.58 | 0.43 | 80.59% | 75.74% | 89.05% | 89.86% | 60.65% | 90.02% |
> |       60%      | 87.16 | 0.38 | 80.62% | 79.17% | 89.80% | 89.73% | 62.09% | 90.60% |
> |       40%      | 84.45 | 0.35 | 79.82% | 78.92% | 88.16% | 89.37% | 65.34% | 88.99% |
>
> Secondly, we admit that there is ambiguity in the current training time measurement. The time measurement varies in the paper due to the change of tasks and the influence of the hardware environment. We strictly measure the time elapsed during each step in the CUDA benchmark mode and take the average per-step time over a large number of steps. The time for data I/O is excluded. The training time of EarlyBERT includes both the searching stage and the efficient training stage. We will clarify how the training time and efficiency is measured and include strictly measured numbers in the updated version.
>
>
> Questions:
>
> Q1: Completely removed layers.
>
> A1: On one hand, from the empirical perspective, the main reason that we choose layerwise pruning for self-attention heads instead of removing the full layer is because we prune the attention heads and FC layers separately. When attention heads are completely pruned, the corresponding FC neurons are not necessarily all pruned. In this situation, we think removing the whole layer will hurt more than layerwise pruning. On the other hand, it is also discovered in Ramsauer et al. (2020) that attention heads in different layers exhibit different behaviors.  This also motivates us to only compare the importance of attention heads within each layer.
>
> Q2: Statistical significance of results in the ablation study (table 3 and 4).
>
> A2: We think the results in the ablation study have fair enough statistical significance because we reported the average accuracy on 4 larger tasks in GLUE (MNLI, QNLI, QQP and SST-2), instead of one run on one task. However, we agree that, as Reviewer 3 also mentioned, the results will be more reliable with more 3-5 runs. This takes some time as it requires a large-scale ablation study. We are running multiple runs of experiments and will report the results soon.

---

### Official Review · AnonReviewer4 · 2020-10-27
**Interesting approach to reduce pre-training approaches for large NLP models**

**Rating:** 6
**Confidence:** 4

**Review:**

The main contribution of this work is to use Early Bird Lottery Tickets to reduce pre-training and fine tuning time for BERT. In order to reduce computation, the authors propose pruning the number of attention heads and neurons in the fully connected layers. They demonstrate that the technique works for BERT-Base and BERT-Large for GLUE and SQUAD tasks.

The work is well-thought through and authors do a good job of explaining their approach and other existing works using lottery tickets.  The usage of lottery tickets during the pre training phase is the biggest strength of the paper since it can result in significant computational savings. The authors perform interesting ablation studies but they could be augmented by a few more experiments (see below). The paper could also be improved by comparison with relevant prior work.

Here are some thoughts and questions that could help improve the paper:

* Experiments in Section 4.2 are only during the fine tuning stage. How do these results compare with prior work for drawing lottery tickets in transfer learning for NLP (Chen et al, Prasanna et. al)? Particularly, it would be good to compare against Chen et al since their work is very relevant.
* How long is the searching stage followed by the efficient training stage? Does this remain the same for all the tasks? Or did it require tuning? It would be interesting to see how soon we can switch from the searching stage to the efficient training stage.
* The paper proposes two different approaches for pruning: pruning attention heads and removing neurons from fully connected layers. It would be interesting to see how the accuracy of the approach changes if only neurons or attention heads are pruned.
* In Section 4.3, the authors discuss reducing training time by reducing training steps for EarlyBERT. I think a similar analysis for BERT would be helpful to understand if the training time can be reduced for the baseline model as well. Also, pre-training techniques have tended to show improvements in downstreams tasks with longer pre-training as the long as the pre-training dataset is large enough. So, perhaps this reduction in pre-training steps might not be applicable for larger pre-training datasets.

Overall, I find the approach interesting and the authors show computational savings in the pre-training models. I recommend accepting the paper.

---

> ### Author Response · Authors · 2020-11-18
> **Response to Reviewer 4**
>
> Thank you for your review and constructive feedback. Below, we provide detailed responses to your questions.
>
> Q1: How do the results in Sec. 4.2 compare with (Chen et al, Prasanna et. al)?
>
> A1: First, we think the transferability of the winning tickets in the context of EarlyBird tickets is an interesting topic to study. And we do observe that the winning tickets drawn from different tasks share some common structure. We leave more careful investigation of the transferability to future work. However, we would like to emphasize that the focus of EarlyBERT is efficient training via structured lottery tickets and their early emergence, which is totally different from (Chen et al, Prasanna et. al). Specifically, Chen et al study the unstructured lottery tickets identified by iterative magnitude pruning. The training cost of IMP could be as high as 10 times of a normal full BERT training, depending on the number of iterations; Prasanna et. al also study structured lottery tickets as in this paper, but they also use the iterative pruning method, as mentioned by Reviewer 2, which makes it hardly applicable to the efficient training scenario.
>
>
> Q2: How long is the searching stage? Does this remain the same for all the tasks? Or did it require tuning?
>
> A2: In the fine-tuning experiments, we run 20% of one full epoch training for the searching stage for all tasks on GLUE and SQuAD. In the pre-training experiments, we run 400 steps of phase 1 for the training stage. This hyperparameter is tuned by observing the convergence of the pruning mask distance (Figure 1). For fine-tuning, we tune this one task and empirically find it generalize well to other tasks. The selection of this hyperparameter shows that the winning tickets emerge early in training, which is one key factor of why EarlyBERT can save the overall training time.
>
>
>
> Q3: It would be interesting to see how the accuracy of the approach changes if only neurons or attention heads are pruned.
>
> A3: This is a good point! We have actually reported these results in Table 4, in the ablation study of pruning ratios and pruning methods for self-attention heads and FC neurons pruning, where we consider pruning self-attention heads only or FC neurons only. We also have such results for pre-training. In summary, the observations are: when we prune self-attention heads or FC neurons only, the performance will be between the baseline BERT models and EarlyBERT that prunes them both, which is expected. We did not present the comparison due to the space limit. Here we provide in the table below a full comparison of the BERT baseline, EarlyBERT with only self-attention heads pruned (4 heads per layer), EarlyBERT with only FC neurons pruned (40% neurons pruned globally) and EarlyBERT with both pruned. We will include them in the Supplementary in the final version.
>
> | Model             | MNLI  | QNLI  | QQP   | SST-2 |
> |-------------------|:-------:|:-------:|:-------:|:-------:|
> | BERT              | 83.16% | 90.59% | 90.34% | 91.70% |
> | EarlyBERT - Heads | 83.58% | 90.33% | 90.41% | 92.09% |
> | EarlyBERT - FC    | 82.47% | 90.21% | 90.38% | 91.74% |
> | EarlyBERT - Both  | 81.81% | 89.18% | 90.06% | 90.71% |
>
>
>
> Q4: Analysis on reducing training steps for BERT.
>
> A4: Thank you for your suggestion on the analysis of training steps reduction during pre-training. We conduct similar experiments as in Figure 3, where we pre-train a BERT-Base model for 100%, 80%, 60% and 40% of the default training steps with all other hyperparameters unchanged. The results are shown below. We can see that on tasks with fairly large dataset (e.g. SQuAD, MNLI, QNLI, QQP and SST-2) the results are relatively consistent: reducing the training steps will hurt the performance of BERT on downstream tasks but with acceptable degradation using 80% and 60% training steps. More obvious accuracy drops will be observed when we reduce the training steps to 40% of the default aggressively. We think this shows that there exists some room for us to reduce the number of training steps by a proper ratio even for BERT. Results on smaller tasks are not very stable, which calls for multiple runs of experiments to exclude the influence of randomness (also suggested by other reviewers). We will include more reliable analysis of the reduction in training steps for BERT and EarlyBERT in the updated version.
>
> | Training Steps | SQuAD | CoLA | MNLI | MRPC | QNLI | QQP | RTE | SST-2 |
> |:--------------|:-----:|:----:|:------:|:------:|:------:|:------:|:------:|:------:|
> |      100%      | 87.60 | 0.45 | 81.40% | 84.07% | 89.86% | 89.80% | 60.29% | 90.48% |
> |       80%      | 87.58 | 0.43 | 80.59% | 75.74% | 89.05% | 89.86% | 60.65% | 90.02% |
> |       60%      | 87.16 | 0.38 | 80.62% | 79.17% | 89.80% | 89.73% | 62.09% | 90.60% |
> |       40%      | 84.45 | 0.35 | 79.82% | 78.92% | 88.16% | 89.37% | 65.34% | 88.99% |

---

### Official Review · AnonReviewer2 · 2020-10-28
**Nice work**

**Rating:** 7
**Confidence:** 3

**Review:**

Summary:
This paper uses the Lottery Ticket Hypothesis to compress BERT. More specifically, they adapt EarlyBird lottery tickets to the BERT setting in order to find winning configurations in early stages of training combine it with structured pruning methods to ensure the resulting network is more efficient to train. The method is a three-stage process: (1) searching - this phase involves training full BERT with some coefficient parameters to learn the mask, (2) drawing - use the mask to “draw a ticket” or select the sub-network to train, (3) training. Experiments show that performance isn’t that much worse when EarlyBERT is used for fine-tuning and for pre-training.

The goal of the paper is to find structured winning tickets for BERT in the early stages of training/fine-tuning.

Strengths:
1. Compressing BERT using the lottery ticket hypothesis has been getting a lot of attention recently, and doing so relatively efficiently is an exciting and interesting contribution.
2. Compared to previous and contemporary work, the combination of EarlyBird lottery tickets (You et al., 2020) to detect tickets early and network slimming (Liu et al., 2017) for structured pruning, is an interesting one. Prasanna et al., (2020) are doing structured pruning too, but via an iterative pruning method.
3. Experiments for both pre-training (the first of its kind) and fine-tuning show that performance does not drop all that much for GLUE and Squad which are the main set of tasks BERT is typically evaluated on.

Weaknesses:
1. The paper isn’t very clear in some places. It starts out explaining things well but then it gets harder to follow the details. Eg: need to add more information on the mask - it’s binarized at some point? The distance metric is still Hadamard like in EarlyBird?
2. (More of a question/nit) Is it possible to also show what happens when a winning ticket for BERT fine-tuning is selected based on the pre-training objective? Chen et al., (2020) showed that these make for better tickets that are performant on many of the downstream tasks. It would be interesting to see if this holds true for EarlyBert and would add another layer of fine-tuning efficiency.

The work isn't terribly novel, but it's still interesting.

Questions and comments:
1. Why does the mask distance diverge for FC in pre-training (Figure 1b)? Does it somehow indicate that the training run is degenerate?
2. The ablation on the regularization parameter didn’t give a clear indication of how important selecting this is. It seems to not be that important, but would using separate values for attention and FC make a difference?
3. It would probably be helpful to expand the table/figure captions, make them a bit more detailed.
4. Curious, why did you only use the largest tasks from GLUE?

---

> ### Author Response · Authors · 2020-11-18
> **Response to Reviewer 2**
>
> Thank you for your positive response and constructive feedback! Below, we provide detailed responses to your questions.
>
> Q1: The mask is binarized at some point? The distance metric is still Hadamard like in EarlyBird?
>
> A1: Yes, your understanding is correct. We use the coefficients c to draw the binary pruning mask. And we use the Hadamard distance between binary masks as in the EarlyBird work. We will make this clear in the revision.
>
>
> Q2: Is it possible to also show what happens when a winning ticket for BERT fine-tuning is selected based on the pre-training objective?
>
> A2: Thank you for this interesting suggestion. We are running this experiment and will report the results here soon.
>
>
> Q3: Why does the mask distance diverge for FC in pre-training (Figure 1b)?
>
> A3: Our conjecture is that when the pre-training runs for enough steps, many parameters will get closer to zero since L1 regularization is used, which imposes constant penalty on all parameters, making the pruning mask less stable. However, as we mentioned in the paper, the influence of the instability of the pruning mask can be avoided by monitoring the mask distance and early stopping.
>
>
> Q4: The ablation on the regularization parameter didn’t give a clear indication of how important selecting this is.
>
> A4: We observe that the selection of the regularization parameter makes a difference for the mask convergence, especially for the pre-training task. We totally agree that it is a good point to use separate values for attention and FC layers. However, it will also make it harder to tune hyperparameters, especially for the expensive pre-training experiments.
>
>
> Q5: Expanding the table/figure captions.
>
> A5: We will add more details to the captions in the final version. Thanks for your suggestion.
>
>
> Q6: Why did you only use the largest tasks from GLUE?
>
> A6: For experiments of fine-tuning, we only use the largest tasks from GLUE because we are more interested in efficient training using the proposed EarlyBERT. When the datasets are too small for some tasks, it makes less points to improve the training efficiency on them.

---

> ### Author Response · Authors · 2020-11-25
> **Updates of the experiments of using pre-training objective during the searching stage**
>
> We have finished the experiments of using the pre-training objective (i.e. MLM, masked language modeling) during the searching stage. The results are summarized in the table below. Our main observations include:
>
> 1. When using the MLM objective for the searching stage, the mask distance for both self-attention heads AND FC neurons converged well and quickly with less than 100 training steps.
> 2. We first apply the global pruning method to the FC neurons because we observed better performance of EarlyBERT with that method. However, while we previously found in EarlyBERT that the latter layers will be pruned more, we observed the opposite phenomenon when using MLM objective --- the former layers are pruned more instead. In terms of accuracy, we observed significant gaps compared to EarlyBERT.
> 3. Based on the above observations, we also applied layerwise pruning for MLM experiments (shown in the last row in the table below). We did see improved accuracy with layerwise pruning but the gaps between EarlyBERT are still large (except on QQP).
>
> We think these results are very interesting (thanks to the great suggestion) and will include them in the updated version of the paper.
>
> |      GLUE Task     |  MNLI  |  QNLI  |   QQP  |  SST-2 |
> |:------------------:|:------:|:------:|:------:|:------:|
> |        BERT        | 83.36% | 90.53% | 90.41% | 91.61% |
> |      EarlyBERT     | 81.97% | 88.68% | 89.26% | 90.48% |
> |   MLM - FC Global  | 78.36% | 84.84% | 88.86% | 88.65% |
> | MLM - FC Layerwise | 79.01% | 86.55% | 89.16% | 88.53% |

---

### Official Review · AnonReviewer3 · 2020-11-02
**Interesting approach but needs some more experiments**

**Rating:** 5
**Confidence:** 3

**Review:**

This paper proposes an approach to sparsifying BERT. What sets this work apart from prior work on model compression is that while prior works attempt to compress a pre-trained BERT, the authors in this work attempt to learn a sparsified BERT for the purpose of speeding up pre-training. The method essentially involves learning an independent bernoulli mask for all BERT heads along with Bernoulli masks for some later intermediate neurons. This model is then trained, along with the masks, for a few epochs. Then, heads/neurons corresponding to a small value of mask are pruned out. This results in a sparser BERT model, leading to faster pre-training.

#### Pros
The paper is well written and the presentation of the contribution is simple and well-motivated.

#### Cons
1. Since one of the main contributions of this work is to make progress on improving the training / inference speed of large transformers, the authors could spend more time going over how the “Time Saved” column is computed. As of now, it seems casual and hand-wavy.
2. Experiment protocol:
  * How were the hyperparameters decided? Could we have uncertainty estimates for all results by reporting mean/std dev. across 3-5 runs (atleast for fine-tuning if doing this for pre-training is too compute intensive). This is especially useful since some of the numbers between EarlyBERT and Random appear very close in Table-1.
  * While a central argument is that most model distillation techniques still require expensive pre-training, it would still be useful to include some of those results in Table-2 since EarlyBERT is comparable to those techniques for the purpose of Table-2.
  * One way to contextualize how much time is saved during pre-training would be to report total time required for fine-tuning+pretraining on the entire glue benchmark. This would involve computing the pre-training time (time to learn BERT parameters on Wikipedia) + total fine-tuning time across all datasets (QQP/CoLA/MNLI etc) considered. This could then be compared against alternatives (such as DistilBert and DeeBERT).
3. Baselines: Experiments on pre-training compare with no baselines. Some possibilities:
  * Using DeeBERT / other early exit approaches at pre-training time.
  * Training a BERT model for some epochs, and then distilling it into a smaller network for the rest of the training.


#### Misc Nitpicks
1. The phrase “structured sparsity’ is used in multiple places, but never defined.
2. You et al. pioneers $\rightarrow$ You et al. pioneer
3. Could Prasanna et al. 2020 be extended to sparsify pre-training?

---

> ### Author Response · Authors · 2020-11-18
> **Response to Reviewer 3 (Continued)**
>
> Q4: Report total time required for fine-tuning+pretraining on the entire glue benchmark.
>
> A4: Pre-training time is observed to dominate the total training time for pre-training + fine-tuning. Hence, we mainly compare the pre-training time. On the other hand, the mentioned alternatives are designed for efficient inference. For example, DistillBERT still requires expensive pre-training. We would also humbly mention that there are not many works for efficient pre-training of BERT models. Thus, we did not provide other alternatives in pre-training experiments.
>
>
> Q5: Experiments on pre-training compare with no baselines.
>
> A5: Thanks for suggesting using DeeBERT as a pre-training baseline, which will be cited in the revision. DeeBERT is a dynamic exiting method, for efficient single sample inference. However, it is not trivial to directly apply it to efficient training. The first but maybe the most important issue is batching. One mini-batch of training samples may have different exit positions. An approach for this is to use zero padding to align all samples with the sample that exits last, which makes the algorithm much less efficient. Secondly, it is also difficult to measure the training time saving due to the dynamic nature of DeeBERT. However, we do agree that distillation with a slightly pre-trained model is a good baseline. We are running the experiment and will report the result soon.
>
>
> Q6: The phrase “structured sparsity’ is used in multiple places, but never defined.
>
> A6: Thank you for pointing out the ambiguity of the term “structured sparsity”. Here, “structured sparsity” is a counterpart term of “unstructured sparsity” used in the original lottery tickets hypothesis (Frankle and Carbin, 2019), where weights in the deep models are zeroed out element-wisely without other structure. In our work, we focus on “structured sparsity” where we will prune out some self-attention head as a whole or some neuron in FC layers (and all weights in the previous & following layer that correspond to that neuron). We will clarify this terminology in the updated version.
>
>
> Q7: You et al. pioneers => You et al. pioneer
>
> A7: We will correct this typo. Thank you!
>
>
> Q8: Could Prasanna et al. 2020 be extended to sparsify pre-training?
>
> A8:  As Reviewer2 mentions, Prasanna et al. (2020) also perform structured pruning but via an iterative pruning method, which is prohibitive for efficient pre-training.
>
> We hope that you can raise your score if you find our answers address your questions. Thank you!

---

> > ### Author Response · Authors · 2020-11-25
> > **Updates about the distillation baseline experiments for pre-training**
> >
> > We thank the reviewer again for suggesting distillation as a baseline for the pre-training experiments. Because this is a new setting and no related work has done similar thing, we find it hard to find a reasonable set of hyperparameters for the distillation, i.e. the coefficients of different losses and the temperature, especially given the cost of one run of the pre-training process and limited time. We failed to train one suggested distilled model that has comparable performance on downstream tasks as EarlyBERT. And we do observe slower training during pre-training in terms of training loss for the distilled model. We will keep searching for the near optimal hyperparameters and report the corresponding results in the final version.

---

> ### Author Response · Authors · 2020-11-18
> **Response to Reviewer 3**
>
> Thanks for your review and constructive suggestions. Below, we provide detailed responses to your concerns.
>
> Q1: The authors could spend more time going over how the “Time Saved” column is computed.
>
> A1: Thank you for your constructive feedback about the training time measurement. Here we clarify the criterion that we used to compute the “Time Saved” column. To get rid of the influence of the hardware environment at our best, we measure time using the CUDA benchmark mode and individually measure the time elapsed during each step. The time for data I/O is excluded. The training time of EarlyBERT includes both the searching stage and the efficient training stage. We compare the total training time of EarlyBERT with a normal BERT model with default fine-tuning settings. Using the above criterion, we observe 42.97% training time saving on the QQP task in GLUE. We will provide the full strict measurement of training time savings on all tasks in fine-tuning and pre-training in the updated version. For a full table of accuracy-time trade-off, please refer to our response to Reviewer 5.
>
>
> Q2: How were the hyperparameters decided? Could we have uncertainty estimates for all results?
>
> A2: For most hyperparameters about training, we follow the hyperparameters that are commonly used in the official implementation of BERT and previous works. We conducted detailed ablation studies in Section 4.2 (Figure 2 and Table 3&4) about the selection of learning rate, number of training epochs, regularization coefficient, pruning ratios and pruning methods (layerwise v.s. global pruning).
>
> We would like to thank the reviewer for the suggestion about repeating the experiments and presenting the uncertainty estimates. As suggested, we conducted 3 runs of fine-tuning experiments for EarlyBERT on QQP and QNLI tasks (due to the limited time) with different random seeds. These experiments correspond to the experiments in Table 2 in the paper. The randomness mainly influences the training data sampling process in the case of fine-tuning as we use a pre-trained BERT model as the initialization. On QQP we have BERT with 90.41% +- 0.12% accuracy and EarlyBERT with 89.25% +- 0.31%. On QNLI we have BERT with 90.53% +- 0.12% accuracy and EarlyBERT with 88.68% +- 0.48% accuracy. In the updated version of this paper, we will present all results with multiple runs and uncertainty estimates.
>
> Specifically for experiments in Table 1, the results of random pruning are already the average of 5 trials with different random seeds for pruning. We can see that the accuracy gaps between EarlyBERT and randomly pruned BERT are significant except on QQP. On QQP, the random BERT model has the best accuracy 90.34% and the worst 89.7%, and the mean and standard deviation is 90.12% +- 0.25%. Therefore, we think the results in Table 1 are still reliable evidence that supports our claim that EarlyBERT finds non-trivial sub-networks indeed. However, we will also conduct multiple runs for the BERT baseline and EarlyBERT to make the results more reliable and convincing in the updated version.
>
>
> Q3: It would be useful to include some of the model distillation results in Table-2.
>
> A3: Thank you for the suggestion. We will include DistillBERT and other fair alternatives in Table 2 in the updated version. However, it is worth noting that EarlyBERT and DistillBERT are more of orthogonal works that could be combined together than competitors. We could extend EarlyBERT by distilling knowledge from a large well-trained teacher to improve the performance.

---

### Official Review · AnonReviewer5 · 2020-11-06
**Unclear experimental results**

**Rating:** 3
**Confidence:** 4

**Review:**

Summary:

The authors propose a technique for reducing the computational requirements of training BERT early in training to reduce the overall amount of resources required.

Pros:

The paper is well written and clear for the most part. The authors do thorough experimental evaluation.

Cons:

I have two primary concerns about the paper and the proposed technique.
1. The positioning of the technique is not entirely clear to me. The authors pitch it as a technique for reducing the training time of BERT and use LayerDrop as a baseline technique that also removes network components. However, it feels like another baseline that should be considered is neural architecture search, which also seeks to automatically find a more efficient model to train. The difference here is that the authors find the model early in the training run, but it seems like the EarlyBERT procedure could be run once and the resulting model architecture could be saved and re-trained like NAS models are.
2. I found the experimental results to be lacking detail and breadth necessary to establish the value of the technique. Firstly, the rough time estimates in Table 2 are very odd given the primary value of the proposed technique is to reduce training time. The accuracy of EarlyBERT is close enough to LayerDrop that accurate training cost numbers are needed to differentiate between the techniques. Secondly, quoting training time reductions over the dense baseline when the EarlyBERT mode does not achieve the same accuracy makes the comparison very difficult to make. This problem shows up quite commonly in the model compression literature [1] and I’d encourage the authors to show full accuracy-training time tradeoff curves so that the training time savings for a given accuracy can be more clearly established. Lastly, I found the use of reduced training epochs in EarlyBERT to be odd because you do not evaluate whether or not this can be done for the baseline models and there isn’t clear evidence as to why your model would be able to do this while others (e.g., DropLayer) cannot. Figure 2 also does not seem to corroborate that higher learning rates can be used with shorter training time to achieve better accuracy. The data in your figure shows that the best learning rate achieves the best model quality independent of the number of training epochs.

References:
1. https://arxiv.org/abs/2003.03033

---

> ### Author Response · Authors · 2020-11-18
> **Response to Reviewer 5**
>
> Thanks for your insightful comments. Below, we provide detailed responses to your concerns.
>
> Q1: It feels like another baseline that should be considered is neural architecture search (NAS).
>
> A1: We believe that you made an inappropriate connection to NAS. First, as you have correctly summarized, the main purpose of the proposed EarlyBERT method is to reduce the total training time of BERT,  **including** the time for the searching stage for the early-bird tickets. The overall training time saving is not only benefited from the efficiency of the pruned model, but also the cheap searching stage. In contrast, NAS requires a heavy searching stage which makes it not a proper baseline here. Second, (early-bird) lottery tickets hypothesis (LTH) is an essentially different approach from NAS. LTH identifies key sparse sub-networks through iterative training protocol as in Frankle and Carbin (2019), or the early-bird protocol used in this work. We will add some discussion, and make this clear in the revision.
>
>
> Q2: The rough time estimates in Table 2 are odd and the comparison with LayerDrop.
>
> A2: First, we would like to highlight the performance gap between EarlyBERT and LayerDrop in Table 2. On the QQP task, EarlyBERT is better than LayerDrop by an absolute 2% accuracy for both BERT-Base and BERT-Large, and 1% on SST-2. We believe this is a non-trivial improvement further considering that EarlyBERT saves more training time than LayerDrop.
>
> Second, thank you for your constructive feedback about the training time measurement. We strictly measure the training time saving of EarlyBERT on the QQP task in GLUE using CUDA benchmark mode. To get rid of the influence of the hardware environment at our best, we individually measure the time elapsed during each step. The time for data I/O is excluded. The training time of EarlyBERT includes both the searching stage and the efficient training stage. We observe that, with 4 self-attention heads pruned in each layer and 40% FC neurons pruned globally, i.e. the setting we used in the paper, EarlyBERT saves 42.97% of the total training time of a full BERT model. We will strictly measure the training time on all tasks in fine-tuning and also pre-training and report them in the updated version.
>
>
> Q3: Please show a full accuracy-training time tradeoff curve.
>
> A3: Thank you for the great suggestion of the full accuracy-training time trade-off curve. We vary the pruning ratios for the FC layers and the number of self-attention heads pruned in each layer in EarlyBERT, fine-tune the models on QQP in GLUE and get the corresponding validation accuracies and training time saved following the protocol above. The tables for the accuracies and training time savings are shown below. We can see that there is an obvious relation between the training time saving and the accuracy --- we can save more by pruning more FC neurons and self-attention heads but suffer larger accuracy drop. Moreover, for the most combinations of these two hyperparameters, the accuracy drop is reasonable (within 1%), which we think also supports the efficacy of EarlyBERT. We will perform the same experiments on all tasks in GLUE and SQuAD for multiple runs to provide more reliable results in the updated version.
>
>
> | Time Saved on QQP          &nbsp;| 3 Heads Pruned     &nbsp;| 4 Heads Pruned    &nbsp;| 5 Heads Pruned     &nbsp; | 6 Heads Pruned |
> |------------------------|----------------|----------------|----------------|----------------|
> | FC Pruning Ratio - 30% |         35.78% |         38.66% |         41.26% |         45.34% |
> | FC Pruning Ratio - 40% |         39.72% |         42.97% |         42.93% |         44.49% |
> | FC Pruning Ratio - 50% |         43.89% |         45.54% |         47.02% |         48.53% |
>
> For comparison, the BERT baseline fine-tuned with default settings has accuracy 90.35% on QQP.
>
> | Accuracies on QQP                 &nbsp;| 3 Heads Pruned            &nbsp;| 4 Heads Pruned           &nbsp;| 5 Heads Pruned        &nbsp; | 6 Heads Pruned |
> |------------------------|----------------|----------------|----------------|----------------|
> | FC Pruning Ratio - 30% |         89.62% |         89.55% |         89.60% |         89.50% |
> | FC Pruning Ratio - 40% |         89.66% |         89.61% |         89.58% |         89.38% |
> | FC Pruning Ratio - 50% |         89.54% |         89.35% |         89.34% |         89.31% |

---

> ### Author Response · Authors · 2020-11-18
> **Response to Reviewer 5 (Continued)**
>
> Q4: I found the use of reduced training epochs in EarlyBERT to be odd.
>
> A4: The rationale of the use of reduced training steps for EarlyBERT echos (Frankle and Carbin, 2019), which observes that early stopping is needed for better performance when re-training the identified winning tickets because much fewer parameters need to be trained. In contrast, although LayerDrop drops layers randomly during training, it still trains all layers and uses them during inference, which might be hurt more with reduced training steps. The results in Table 2 show that LayerDrop is inferior to EarlyBERT even with full training steps.
>
>
> Q5: Figure 2 also does not seem to corroborate that higher learning rates can be used with shorter training time to achieve better accuracy.
>
> A5: Figure 2 is presented as an ablation study to find the best combination of learning rate and reduced training epochs, in which we achieve the best performance with 4e-5 learning rate and 2 epochs of training. We then stick to this optimal setting in the fine-tuning experiments. Sorry for the confusion, and we will make this clear in the revision.
>
> We hope that you can raise your score if you find our answers address your questions. Thank you!

---

> ### Comment · AnonReviewer5 · 2020-11-24
> **Reviewer Response 1**
>
> Thank you to the authors for your responses. After reading your rebuttal, I have the following remaining concerns
>
> 1. I disagree that NAS is an inappropriate baseline. For any practical application of DNNs there is a cost F of finding the model you want to train and then a cost T of training the model for deployment. In practice, once an architecture is found it is re-trained many times (e.g., over time as more data is collected and the dataset is updated). Thus, the total training cost is something like F + a * T, where 'a' is the number of re-trainings. NAS has a high F, but this can be amortized for deployments where 'a' is high. The proposed approach has a low F, but likely a higher T than a more complex search process like NAS. I understand that it's not practical to require the author's to compare to every alternative technique for finding a more efficient model to train, but it's worth acknowledging that for applications where 'a' is high (likely most real deployments) the efficiency gains over more complex techniques like NAS are likely to be negligible (or negative, assuming NAS produces more efficient architectures).
> 2. The additional runtime-accuracy tradeoff numbers are helpful, but I do not think they are sufficient to establish the value of the proposed technique. I would like to see comparisons with more baselines like NAS, smaller BERT models, and more recent and more efficient architectures.
>
> For the time being I am maintaining my score as is.

---

> > ### Author Response · Authors · 2020-11-25
> > **Clarification about the comparison with NAS**
> >
> > Thank you for your response. We respectfully but firmly disagree with your argument about NAS. We present our clarifications below and hope they can help you to better position our work.
> >
> > ### Argument about EarlyBERT and NAS
> >
> > Firstly, we would like to emphasize that the motivation and the goal of this paper is about one-time efficient training of a BERT type model, with a pre-defined architecture and the data set. In this “one-time” training setting, the cost of NAS will not be comparable with the proposed method at all.
> >
> > Secondly, we do not consider the scenario where we need to re-train the network because of the shift of data distribution (due to more data collected or data set update, as you mentioned as examples), which is completely outside the scope of this work. Moreover, it is also not clear that if the “efficient” model searched by NAS will be optimal when the dataset changes, especially when the number of re-training “a” is high, which means that there are many changes in the data set and thus the shift of data distribution could be very significant. It is not any trivial intuition about the transferability of the searched architecture. In that case, a re-search process could also be required, which also significantly increases the cost of NAS, too.
> >
> > Thirdly, although it is not empirically verified yet, we see no technical difficulty that stops us from applying EarlyBERT to the searched architecture by NAS. We argue that these two techniques are totally different and are orthogonal to some extent, that could be combined together.
> >
> > ### Comparison with NAS, smaller BERT models, and more recent efficient models
> >
> > For NAS, please refer to our argument above in this response.
> >
> > We would like to humbly remind you that this paper is the very first work about efficient training of BERT type models in both pre-training and fine-tuning. Almost all efficient BERT models are focused on inference time efficiency, and some of them will induce expensive training costs and thus are not suitable for efficient training. For example, AnonReviewer3 also mentioned comparison with DistilBERT and DeeBERT. However, DistilBERT has a heavy general-purpose distillation training process on the Mask Language Modeling task and then performs task-specific fine-tuning on downstream tasks, which is much more expensive in terms of the training cost than EarlyBERT. DeeBERT is also not suitable for efficient training because the dynamic routing in DeeBERT is efficient for one sample inference but can hardly improve training efficiency in modern batched training manner (refer to our response to AnonReviewer3). It is also worth noting that EarlyBERT is actually orthogonal to DistilBERT as we could extend EarlyBERT by distilling knowledge from a large well-trained teacher to improve the performance.
> >
> > However, we will include the comparison with DistilBERT and other fair alternatives as baselines in the updated version.

---

### Decision · Program_Chairs · 2021-01-07
**Final Decision**

**Decision:**

Reject

**Comment:**

It is important to develop efficient training methods for BERT like models since they have been widely used in real-world natural language processing tasks. The proposed approach is interesting. It speeds up BERT training via identifying lottery tickets in the early stage of training. We agree with the authors's rebuttal that autoML is not that related to the work here. Our main concern on this work  is its worse-than-BERT performance showed in Table 2. The performance gap is significant. Sufficiently more training steps would fill the performance gap but the proposed method may have no advantage any more over the normal training procedures. To make this work more convincing, we would like to suggest to include experiments on comparing different methods under similar prediction performance.  In addition, since the main claim of this work is for training efficiency,  it will be helpful to show the advantage of this method by directly presenting the training curves/ results of different methods.   Overall this paper is pretty much on the boundary. We encourage the authors to resubmit this work once these issues are well resolved.